# 6-(Methylsulfonyl) Hexyl Isothiocyanate: A Chemopreventive Agent Inducing Autophagy in Leukemia Cell Lines

**DOI:** 10.3390/biom12101485

**Published:** 2022-10-14

**Authors:** Veronica Cocchi, Beatriz Jávega, Sofia Gasperini, José-Enrique O’Connor, Monia Lenzi, Patrizia Hrelia

**Affiliations:** 1Department of Pharmacy and Biotechnology, Alma Mater Studiorum University of Bologna, 40127 Bologna, Italy; 2Laboratory of Cytomics, Joint Research Unit CIPF-UVEG, Department of Biochemistry and Molecular Biology, University of Valencia, 46010 Valencia, Spain

**Keywords:** 6-(Methylsulfonyl) hexyl isothiocyanate, 6-MITC, autophagy, ROS, leukemia cell lines, Jurkat cells, HL-60 cells, chemoprevention, flow cytometry

## Abstract

Autophagy is a fundamental catabolic process of cellular survival. The role of autophagy in cancer is highly complex: in the early stages of neoplastic transformation, it can act as a tumor suppressor avoiding the accumulation of proteins, damaged organelles, and reactive oxygen species (ROS), while during the advanced stages of cancer, autophagy is exploited by cancer cells to survive under starvation. 6-(Methylsulfonyl) hexyl isothiocyanate (6-MITC) is the most interesting compound in the *Wasabia Japonica* rizhome. Recently, we proved its ability to induce cytotoxic, cytostatic, and cell differentiation effects on leukemic cell lines and its antimutagenic activity on TK6 cells. In the current study, to further define its chemopreventive profile, Jurkat and HL-60 cells were treated with 6-MITC for 24 h. The modulation of the autophagic process and the involvement of ROS levels as a possible trigger mechanisms were analyzed by flow cytometry. We found that 6-MITC induced autophagy in Jurkat and HL-60 cells at the highest concentration tested and increased ROS intracellular levels in a dose-dependent manner. Our results implement available data to support 6-MITC as an attractive potential chemopreventive agent.

## 1. Introduction

The term autophagy, from the Greek *autòs* “oneself” and *phagéin* “to eat”, is a homeostatic, catabolic process highly conserved in all eukaryotes [1,2]. Three main types of autophagy have been characterized in mammalian cells: microautophagy, Chaperone-Mediated Autophagy (CMA), and macroautophagy; those are morphologically different processes, but all three end with the degradation and recycling of cellular components within lysosomes [3,4,5]. Macroautophagy is the most studied mechanism, referred to when speaking more generally about autophagy. It is a finely regulated process. In fact, in normal conditions, it is found at very low levels just to prevent the gradual accumulation of damaged proteins and organelles, which over time would be toxic to the cell [4,6]. However, in stressful situations, such as nutrient and energy deficiency or hypoxic conditions, autophagy is induced to provide an alternative resource of metabolic substrates indispensable for cellular survival. Furthermore, autophagy performs multiple functions: it is involved in embryonic development, cellular differentiation and proliferation, aging, and innate immunity. It also defends the organism from infection by viruses and bacteria through intracellular degradation [4,7,8]. It is, therefore, mainly a cytoprotective mechanism; however, dysfunctions of the autophagic system are associated with numerous pathologies such as neurodegenerative diseases, cardiomyopathies, cellular aging, metabolic dysfunctions, and cancer [4,8,9]. Macroautophagy is characterized by the formation of an autophagosome. This double membrane structure sequesters cytoplasmic proteins, mitochondria, endoplasmic reticulum, and ribosomes, fusing with a lysosome resulting in the formation of an autolysosome. Lysosomal enzymes thus degrade the autophagic cargos and the damaged materials are recycled [10,11,12].

Since autophagy is a multi-step process, its biomarkers include different categories: biomarkers of autophagosomes, lysosomes, and autophagic substrate biomarkers [12]. The primary autophagy biomarker is the microtubule-associated protein LC3II or LC3B, formed from the cleavage and lipidation of the cytosolic protein LC3I and then incorporated into the autophagosome inner leaflet of the membrane in the form of LC3B or LC3II [13]. Other biomarkers for macroautophagy include various autophagy-related (Atg) proteins and autophagy modulators, such as Atg5–Atg12, Atg16L, Atg9, BECN1/Beclin1/Vps30/Atg6, Atg14/Bakor, DRAM1, and ZFYVE1/DFCP1 or substrates of macroautophagy like p62 [12]. Another method for investigating the autophagic process involves acidotropic dyes, such as monodansylcadaverine, acridine orange, Neutral Red, LysoSensor Blue, and LysoTracker Red, which accumulate in acidified vesicular compartments of the cell and, therefore can label lysosomes. However, this method cannot always distinguish between endosomes, amphisomes, lysosomes, and other acidified organelles. So, in some cases, these methods have been gradually replaced, but for some of these dyes, the selectivity has been improved [14,15].

Since Yoshinori Oshumi received the Nobel Prize in Medicine in 2016, the study of autophagy is becoming increasingly attractive. The discoveries of this Japanese scientist made it possible to define the mechanism underlying the autophagic process and the many related pathologies, including cancer [16]. The maintenance of cellular homeostasis through autophagy is essential to cancer prevention; nevertheless, this mechanism represents “a double-edged sword” since it is not only involved in suppressing cell survival in the tumor environment, but it is also capable of promoting it. This paradox can be better understood by analyzing the role of autophagy during tumor progression [17,18]. In the early stages of tumor transformation, autophagy represents a defense mechanism against neoplastic growth. The elimination of damaged mitochondria prevents the accumulation of reactive oxygen species (ROS), which are now recognized as the primary source of oxidative stress in the cell. Consequently, inefficiency in this process at an early stage of cancerogenesis leads to chronic oxidative stress, accumulation of damaged mitochondria, tissue damage, and inflammation: events that jointly promote tumor initiation. Conversely, in later stages, autophagy represents an adaptive response of cancer cells. They can exploit this mechanism as a response to cellular stress and/or increased metabolic demands due to rapid cell proliferation, thus promoting tumor growth and chemotherapeutic resistance. In this case, inhibiting autophagy could be an essential strategy [19,20].

Despite the numerous therapeutic interventions available (chemotherapy, surgical removal, radiotherapy), cancer is still today the second leading cause of death; for this reason, new anticancer approaches are fundamental [21,22].

Preclinical and clinical studies associate phytochemicals-rich diets with a lower risk of developing chronic degenerative diseases, including cancer [22,23]. Phytochemicals are compounds naturally present in plants and include a highly heterogeneous set of chemicals: flavonoids, alkaloids, tannins, triterpenes, tocopherols, phenols, flavonoids, and isothiocyanates (ITCs) [23,24,25]. Thanks to their anti-inflammatory, antioxidant, antiproliferative, and proapoptotic properties, the phytochemicals are extremely noteworthy molecules able to counteract the incidence and mortality of cancer [26,27,28,29].

Among ITCs, 6-(methylsulfinyl) hexyl isothiocyanate (6-MITC), the main bioactive compound present in the *Wasabia japonica* rhizome, has sparked great interest among researchers. Indeed, this ITC has been studied in several models. It inhibits several inflammatory factors such as COX-2, iNOS, and inflammatory cytokines at the transcription factor/promoter levels and has an anti-inflammatory and anti-oxidant effect in in vitro and in vivo Alzheimer’s Disease models [30,31]. Other 6-MITC’s effects concern its antioxidant effect [32,33,34,35], antimicrobial activity [36,37], cytoprotective effect against ethanol- and acetaldehyde-induced cytotoxicity on HEPG2 cells [38], and antiplatelet effect [39]. Regarding the chemopreventive and antitumoral potential of this ITC, several studies proved an anticancer potential on different in vitro models, e.g., on a murine hepatoma cell line and on human breast, melanoma, pancreatic, colorectal, oral, hepatoblastoma, and epidermoid cancer cell lines [32,33,39,40,41,42,43,44,45,46,47,48,49,50]. They also showed an in vivo antitumoral potential against breast cancer and pulmonary metastasis in mice [45,51].

In addition, in two recent publications, we have proved that 6-MITC elicits cytotoxic and cytostatic effects, induces cell differentiation on the leukemic cell lines Jurkat and HL-60, and exerts antimutagenic activity on TK6 cells [52,53]. Overall, these results demonstrated the capacity of 6-MITC to modulate several mechanisms supporting its antitumor activity. In parallel to our studies, Wu et al. demonstrated on human chronic myelogenous leukemia K562 cells that 6-MITC treatment elevates the levels of acidic vesicular organelles compared to controls [50]. Therefore, starting from this first outcome present in the literature, to expand upon our previous studies and better define the chemopreventive profile of 6-MITC, we evaluated the modulation of the autophagic process and the generation of intracellular ROS in Jurkat and HL-60 cells by flow cytometry (FCM), as a possible trigger mechanism for the autophagy.

## 2. Materials and Methods

### 2.1. Reagents

Fetal Bovine Serum (FBS), L-Glutamine (L-GLU), Penicillin-Streptomycin solution (PS), Roswell Park Memorial Institute (RPMI) 1640 medium, 2′-7′-dichlorodihydrofluorescein diacetate (DCFH_2_-DA); 4′,6-diamidino-2-phenylindole (DAPI); N-acetylcysteine (NAC), hydrogen peroxide (H_2_O_2_) (all purchased from Merck, Darmstadt, Germany), CYTO-ID^®^ Autophagy Detection Kit 2.0 (purchased from Enzo Life Science, Farmingdale, New York, NY, USA), Guava Nexin Reagent (containing 7-aminoactinomycin (7-AAD) and Annexin-V-PE) (purchased from Luminex Corporation, Austin, TX, USA).

### 2.2. 6-MITC

6-MITC was purchased from Abcam, Cambridge, UK. The purity of ITC was >98%. The ITC was dissolved in DMSO up to 97.39 mM stock solution and stored in the dark at −20 °C.

### 2.3. Cell Culture and Treatment

#### 2.3.1. Jurkat

Jurkat cells (acute T lymphoblastic leukemia) were grown at 37 °C, and 5% CO_2_ in RPMI-1640 supplemented with 10% FBS, 1% PS, and 1% L-GLU. To maintain exponential growth, the cultures were divided every three days in fresh medium, and the cell density did not exceed the critical value of 3 × 10^6^ cells/mL.

In all the experiments, aliquots of 3.75 × 10^5^ of Jurkat cells were treated with increasing concentrations of 6-MITC in the range of 0–8 µM. The autophagy induction and the level of ROS were measured after 24 h of treatment.

#### 2.3.2. HL-60

HL-60 cells (acute promyelocytic leukemia) were grown at 37 °C, and 5% CO_2_ in RPMI-1640 supplemented with 20% FBS, 1% PS, and 1% L-GLU. To maintain exponential growth and reduce spontaneous differentiation, the cultures were divided every three days in fresh medium, and the cell density did not exceed the critical value of 1 × 10^6^ cells/mL.

In all the experiments, aliquots of 1.25 × 10^5^ of HL-60 cells were treated with increasing concentrations of 6-MITC in the range of 0–16 µM.

The autophagy induction and the level of ROS were measured after 24 h of treatment.

### 2.4. Flow Cytometry

All FCM analyses reported below were performed using a Gallios 3L 10C flow cytometer equipped with three lasers operating at 488 nm, 633 nm, and 405 nm (Beckman Coulter, Brea, CA, USA), a Cytomics FC500 flow cytometer equipped with two lasers operating at 488 nm and 633 nm (Beckman Coulter, Brea, CA, USA) or a Guava easyCyte 5HT flow cytometer equipped with a class IIIb laser operating at 488 nm (Luminex Corporation, Austin, TX, USA).

#### 2.4.1. Autophagy Analysis

To evaluate the potential pro-autophagic activity of 6-MITC, the CYTO-ID^®^ Autophagy Detection Kit 2.0 By Enzo Life Sciences was employed. The assay provides a rapid, specific, and quantitative approach for monitoring the autophagic process at the cellular level, validated under a wide range of conditions known to modulate autophagy pathways [54].

The analysis was performed by FCM.

Briefly, at the end of the treatment time, 1 × 10^5^ to 1 × 10^6^ cells were washed in cell culture medium and stained with CYTO-ID Green stain solution for 30 min at 37 °C in the dark.

DAPI was used to exclude necrotic cells during data analysis. The results were expressed as a percentage of autophagic cells in treated cultures compared to those in the concurrent negative control cultures.

500 nM Rapamycin (RAP) was used as a positive control since it is a well-known inducer of autophagy. 10 µM Chloroquine (CLQ) was used as an inhibitor of the autophagic vesicles degradation since it acts as a lysosomal inhibitor by raising the lysosomal pH and ultimately inhibits the fusion between autophagosomes and lysosomes, thus preventing the maturation of autophagosomes into autolysosomes; as a result, CLQ blocks a late step of autophagy leading to an accumulation of autophagic vesicles in the cell [15].

5 mM NAC was used as an antioxidant agent in association with 6-MITC to evaluate oxidative stress as a possible mechanism underlying the pro-autophagic effect.

#### 2.4.2. ROS Analysis

The analysis of ROS levels was performed by an FCM assay using DCFH_2_-DA. Briefly, at the end of the treatment, 5 × 10^4^ to 3 × 10^5^ cells were stained at 37 °C in the dark for 20 min with DCFH2-DA, a cell-permeant non-fluorescent fluorogenic substrate. The acetate groups of the probe are removed by intracellular esterases, yielding DCFH_2_, which is retained within the cell. ROS-mediated oxidation subsequently generates green-fluorescent 2′,7′-dichlorofluorescein (DCF) [55].

The fluorescence intensity of DCF measured in treated cultures was normalized on that recorded in the untreated control cultures, equal to 1 and expressed as ROS fold increase. DAPI was used to exclude necrotic cells during data analysis. We used 100 µM H_2_O_2_ as a positive control.

### 2.5. Statistical Analysis

Each concentration of the test chemical was tested in triplicate at all the experimental conditions. All analyses were repeated three times. Autophagic cell percentage and ROS fold increase were expressed as mean ± SEM. At all experimental conditions, more than three groups of matched data were compared, so statistical significance was analyzed by one-way repeated measures ANOVA, followed by Dunnett as post-tests to compare all treated groups to the control group. We considered the difference between means statistically significant if the *p* value < 0.05. We used Prism Software 4 (GraphPad Software, San Diego, CA, USA).

## 3. Results

### 3.1. Autophagy Analysis

To verify if 6-MITC can modulate the autophagic process, Jurkat and HL-60 cells were treated for 24 h at the concentrations 4, 8 μM and 8, 16 μM respectively. These concentrations were lesser than or equal to IC_50_ for 6-MITC in Jurkat and HL-60 based on the results obtained in our previous work, where the IC_50_ value calculated by interpolation was 8.65 μM for Jurkat cells and 16 μM for HL-60 cells [52].

RAP, an inhibitor of mTOR (i.e., a stimulator of autophagy), was used as a positive control.

As shown in Figure 1, in Jurkat cells, 6-MITC caused a statistically significant increase in the percentage of autophagic vesicles only at the concentration 8 μM (24.7% vs. 2.9% 6-MITC 0 μM). In comparison, a visible but not statistically significant increase was observed at 4 μM (11.1% vs. 2.9% 6-MITC 0 μM).

In tumor cell lines, the rapid formation and degradation of autophagosomes (i.e., autophagic vesicles) by lysosomes may lead to a low signal. In these cases, induction of autophagic flux is best observed through the accumulation of autophagic vesicles, achieved by using a lysosomal inhibitor, such as CLQ, that prevents the removal of these vesicles [56]. For these reasons, leukemia cell lines were also co-treated with 6-MITC in association with CLQ to better quantify the autophagic cells.

The co-treatment with CLQ showed a clear and statistically significant increase in the percentage of autophagosomes at both concentrations tested. In particular, an increase of 3.6 and 3 times was observed in the cultures simultaneously treated with 6-MITC 4 μM or 8 μM and CLQ compared to that recorded in cultures treated with 6-MITC alone (39.8% vs. 11.1% for 6-MITC 4 μM and 76.8% vs. 24.7% for 6-MITC 8 μM) (Figure 1A–E).

Conversely, the treatment with 6-MITC on HL-60 cells showed no statistically significant increase in the percentage of autophagic vesicles at either concentration tested (8 and 16 μM) even though a doubling was observed at the highest concentration tested (12.3% vs. 5.2% 0 μM).

Also, in this case, when 6-MITC 8 and 16 μM were associated with CLQ, a statistically significant increase in the percentage of autophagic vesicles was observed. More specifically, the percentage of autophagic vesicles for 6-MITC 8 μM + CLQ association was, on average, triple that of 6-MITC alone. In contrast, for 6-MITC 16 μM + CLQ association, a six-fold increase was measured (25.4% vs. 7.6% 6-MITC 8 μM and 78.56% vs. 12.3% 6-MITC 16 μM) (Figure 2A–E).

To evaluate whether autophagy was one of the significant contributors to the cytotoxic effect of 6-MITC, Jurkat and HL-60 cells were co-treated for 24 h with 6-MITC, and the lysosomal inhibitor CLQ, and cytotoxicity was measured using the double staining 7-AAD/Annexin-V-PE to distinguish viable, necrotic and apoptotic cells. The results did not permit evidence of any significant change in the cytotoxic profile of 6-MITC.

### 3.2. ROS Analysis

To evaluate a possible trigger mechanism behind 6-MITC’s capability to induce autophagy in leukemia cells, its potential modulatory effect on ROS levels was investigated in both cell lines. Jurkat and HL-60 cells were treated with concentrations ≤ IC_50_ for 24 h.

As shown in Figure 3, both in Jurkat and HL-60 cells, a statistically significant ROS fold increase was observed (Figure 3A,B).

In particular, on Jurkat cells, a statistically significant increase in ROS levels was detected in the cultures treated from 2 to 8 μM compared to the concurrent negative control cultures (Figure 3A). In contrast, on HL-60 cells, a statistically significant increase in ROS levels was observed only at the two highest concentrations tested (Figure 3B).

To corroborate the hypothesis that oxidative stress may be involved in the pro-autophagic effect of 6-MITC, both cell lines were co-treated with the 6-MITC and the antioxidant NAC. The results showed a decrease in the CYTO-ID mean green fluorescence of the cell population cotreated with 6-MITC and NAC compared to that of the cell population treated only with 6-MITC (Appendix A).

## 4. Discussion

Preclinical, clinical, and epidemiological studies support a close correlation between using natural compounds and the lower risk of developing several types of cancer [57,58].

Phytochemicals are sparking increased interest in chemoprevention due to their pleiotropic activity, i.e., the ability to act through multiple mechanisms. For example, some molecules act early, preventing the activation of pro-carcinogens or favoring the elimination and detoxification of carcinogens through the modulation of biotransformation enzymes; others act on already transformed cells, stimulating apoptosis, arresting/slowing their proliferation or inducing cytodifferentiation, which represent three fundamental mechanisms of chemoprevention. Among the phytochemicals with potential chemopreventive activity, great attention has been paid for a long time to ITCs, the main bioactive compounds present in cruciferous vegetables that can modulate a large number of cancer-related targets, including cytochrome P450 enzymes, proteins involved in the antioxidant response, tumorigenesis, apoptosis, the cell cycle and metastasis [20,21,52]. One of the best-known ITC is Sulforaphane, on which many studies are available in the literature [59,60,61]. Still, in the last few years, another ITC has stimulated the interest of researchers as a chemopreventive agent, 6-MITC, the most interesting bioactive compound present in high concentrations in *Wasabia Japonica* rhizome (better known as Wasabi).

Research previously conducted in our laboratories proved the capacity of this ITC to interact with different cellular and molecular targets critical in neoplastic development, supporting its antitumoral activity [52,53]. In particular, in two other leukemia cell lines (Jurkat and HL-60), 6-MITC has been demonstrated to be capable of cytotoxic, cytostatic, and cytodifferentiation effects. The analysis of the specific mechanism of cell death (apoptosis and necrosis) demonstrated 6-MITC’s ability to induce apoptosis in a dose- and time-dependent manner in both cell lines tested.

In the same study, 6-MITC also exhibited antiproliferative effects in both cell lines, as evidenced by the distribution of cells in the different phases of the cell cycle, in fact, it limited Jurkat cell replication by slowing down the cell cycle, causing a resultant reduction in the percentage of S phase cells after 24 h of treatment, and it be able to induce a potent inhibitory effect on HL-60 cell proliferation, resulting in a blockage of the cell cycle’s progression in the G1 phase, statistically significant after 48 h of treatment.

Moreover, by measuring the expression levels of CD-14 and CD-15 (membrane proteins characteristic of macrophages and granulocytes, respectively), 6-MITC demonstrated its ability to induce cytodifferentiation of promyelocytic cells into both macrophage and granulocyte phenotypes [52].

Besides the capacity to interact with cellular and molecular targets, which is crucial in cancer development, the study and the identification of compounds capable of counteracting genotoxicity are recognized to be of great interest in the field of chemoprevention [62]. If a mutation occurs in a somatic cell, it may lead to premature aging, damage to the immune system, and promote the development of chronic degenerative diseases, such as cancer [63].

For this reason, we decided to perform another study by analyzing on TK6 cells the antigenotoxic ability of 6-MITC against two known genotoxic agents characterized by a different mechanism of action, i.e., the clastogen Mitomycin C (MMC) and the aneuploidogen Vinblastine (VINB). On the one hand, 6-MITC was unable to counteract MMC genotoxic effect. Still, on the other hand, it displayed a potential antigenotoxic activity against the aneuploidogen Vinblastine, which acts on cellular mitosis by preventing tubulin polymerization and consequently inhibits the microtubule aggregation [64]. This result seems to suggest that the ITC tested cannot counteract a structural DNA damage but that it can intervene in the mitotic spindle formation or at the chromosomal segregation time [53].

In parallel to our research, other studies have been published supporting the antitumoral potential of 6-MITC [32,33,39,40,41,42,43,44,45,46,47,48,49,50,51].

In particular, Wu et al. demonstrated on human chronic myelogenous leukemia K562 cells that 6-MITC treatment elevates the levels of acidic vesicular organelles compared to controls [50]. In addition, Hsuan et al. showed in the human colon cancer Colo 205 cell line that the whole wasabi extract stimulates the activation of the autophagic process [46].

Autophagy is an event increasingly investigated and notoriously involved in the cancerogenic process. In fact, as is apoptosis or cellular senescence, it is a fundamental process that may play a positive or detrimental role for the organism [65,66]. It plays an extremely complex and controversial role in cancer development. In particular, in the early stages of neoplastic transformation, autophagy can act as a tumor suppressor, preventing the accumulation of damaged proteins, organelles, and ROS that promote DNA mutations. On the contrary, it is well known that cancer cells reprogram their metabolic machinery to avoid cell death; when the tumor environment is hypoxic and nutrient-poor, autophagy helps transformed cells to adapt to changing conditions by preventing their apoptotic death [66].

Therefore, the discovery of this dual role of autophagy in cancer has led to the development of new anticancer strategies: inducing autophagy upstream, pushing the cell to form autophagic vesicles, and simultaneously blocking the digestive process downstream, eventually leading to the cell’s inevitable death [67].

In this context, the idea was born to investigate the role of autophagy in the chemopreventive potential demonstrated by 6-MITC on Jurkat and HL-60 leukemia cells, used as models in our previous studies. The CYTO-ID staining allowed us to demonstrate 6-MITC’s ability to induce the formation of autophagosomes in both cell lines tested. One of the conventional ways to measure the increased numbers of autophagosomes is to monitor autophagic activity. However, autophagosome formation is an intermediate stage in the whole dynamic autophagy process. In some cell lines (e.g., Jurkat cells), the formation and rapid degradation of autophagic vesicles by lysosomes may result in a low signal. In these cases, induction of autophagic flux is best visualized by inhibiting lysosomal function and preventing the removal of vesicles [54]. A typical lysosomal inhibitor is CLQ. This antimalarial drug enters the lysosome, where the low pH converts it into its protonated form. Due to the positive charge of this compound, it is no longer able to exit and remains trapped within the organelle. The accumulation of the protonated form inside the lysosome leads to a reduction in its acidity and consequently, to a decrease in lysosomal function [68].

We, therefore, considered it appropriate to evaluate the association of 6-MITC with CLQ, which allowed us to measure a sharp increase in the percentage of autophagic vesicles. The results are extremely interesting as the combination of 6-MITC with this lysosomal inhibitor represents a potential strategy to enhance the anti-leukemia efficacy of 6-MITC. Several studies show that the simultaneous stimulation and blockage of the autophagic process increases apoptosis levels, thus stimulating the elimination of tumor cells [69,70,71].

The study continued by analyzing ROS induction by 6-MITC to identify a possible mechanism underlying the autophagic activity demonstrated. It is known that oxidative stress can activate the autophagic process [5,19]. Our results highlighted a statistically significant increase in ROS levels for both leukemia cell lines. Then, we carried out an additional test in which we simultaneously treated both Jurkat and HL-60 cells with 6-MITC in association with the well-known antioxidant NAC for 24 h. At the end of the co-treatment time, we stained the cells with CYTO-ID and observed a decrease in the mean intensity of its green fluorescence.

This result is observable in both cell lines, albeit more markedly in HL60 cells. It allows us to hypothesize a possible involvement of the modulation of ROS levels in the stimulation of the autophagic process by 6-MITC. This outcome seems consistent with previous findings that showed that on SW872 cells, the treatment with the protease inhibitor atazanavir determined an increase in ROS levels (mitochondrial superoxide) and autophagy [72].

On the other hand, it must be taken into account that this ITC is reported in the literature as endowed with an antioxidant capacity [32,33,34,35] which might seem contrary to what we observed in the present study.

However, a first consideration concerns that the cells used in our study are tumoral and, therefore present alterations in metabolism and antioxidant systems that favor an increase in ROS production. Moreover, the molecular characteristics of cancer cells make them more sensitive to stimuli that determine a further increase in ROS. Exceptionally high ROS levels lead to a condition of oxidative stress with consequent cell death [73]. Many anticancer drugs exploit this aspect. For this reason, 6-MITC could, at particular doses and treatment times, cause an increase in ROS levels that force the cell to undergo autophagy and/or apoptosis. Moreover, a quick analysis of the available data already allows us to ascertain that in most of the studies, the levels of gene and/or protein expression of antioxidant enzymes [32,33,34,35] or the protective effect against oxidative stress induced by known pro-oxidants in terms of improving cell survival [34] were analyzed. Furthermore, the results were obtained on different cell types treated with different concentrations of 6-MITC for generally short treatment times or, in any case, for less than 24 h as we did. A comparison between such different studies is, therefore difficult and inconclusive. To deepen this aspect goes beyond the scope of the present work, and the results we have obtained so far are highly preliminary, but they can provide a useful additional starting point to reflect on future research planning.

A final consideration must be made about apoptosis-autophagy interconnection since 6-MITC was previously demonstrated to have a pro-apoptotic effect and is now a pro-autophagic effect.

The quantification of the autophagy contribution to cytotoxicity is not an easy task.

To clarify the relationship between apoptosis and autophagy in terms of cell death induction and answer the question if autophagy was one of the significant contributors of the cytotoxicity of 6-MITC, we evaluated the cytotoxicity of 6-MITC against Jurkat and HL-60, combining it with the lysosomal inhibitor CLQ and evaluating cytotoxicity in terms of viable, necrotic and apoptotic cells. However, the result obtained did not permit evidence of any significant change in the cytotoxic profile of 6-MITC.

The proteins that control apoptosis and autophagy regulation and execution are closely related to [74,75].

The apoptosis induced by 6 MITC in Jurkat and HL-60 cells triggers extrinsic pathways with a reduction of BAX/BCL-2 ratio: BAX levels remained unaltered in both control and treated cultures, while BCL-2 expression was upregulated in the treated cultures [52].

BCL-2 is certainly a component of the apoptotic machinery; however, it can also regulate autophagy via interaction with autophagy proteins. In fact, BCL-2 is typically bound to Beclin 1, a protein involved in the early stages of the autophagic process, preventing the interaction with PI3K-III complex, the main actor in phagophore formation, and consequently inhibiting autophagy. Under autophagy-inducing cell stress, instead, BCL−2 dissociates from Beclin 1 and activates the autophagic process [4,75]. Therefore, the high levels of BCL-2 and the induction of the autophagic process demonstrate a close correlation between the two mechanisms. Confirming this hypothesis would require analyzing many autophagy and apoptosis-related proteins to more clearly define the interconnection of the two death mechanisms.

In conclusion, our results implement available data to support 6-MITC as an attractive potential chemopreventive agent.

## Figures and Tables

**Figure 1 biomolecules-12-01485-f001:**
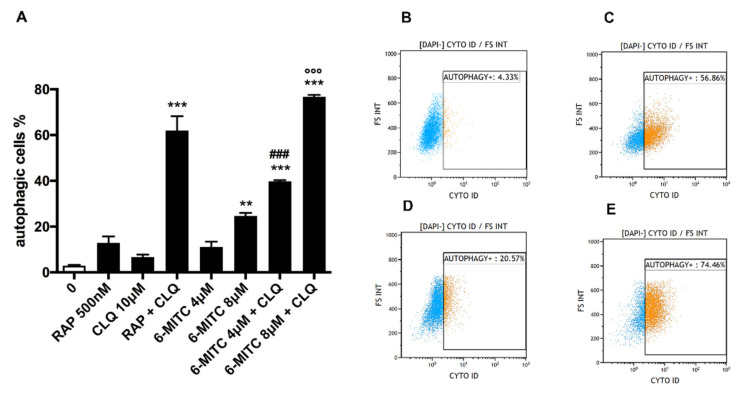
Percentage of autophagic cells in Jurkat cells after 24 h treatment with 6-MITC and/or positive control RAP and/or lysosomal inhibitor CLQ at the indicated concentrations compared to the concurrent negative control cultures (0 μM) (**A**). Each bar represents the mean ± SEM of three independent experiments followed by the Dunnet post-test. ** *p* < 0.01 vs. 0 μM; *** *p* < 0.001 vs. 0 μM; ### *p* < 0.001 vs. 6-MITC 4 μM; °°° *p* < 0.001 vs. 6-MITC 8 μM. Representative FCM plots of 6-MITC 0 μM (**B**), RAP + CLQ (**C**), 6-MITC 8 μM (**D**) and 6-MITC 8 μM + CLQ (**E**).

**Figure 2 biomolecules-12-01485-f002:**
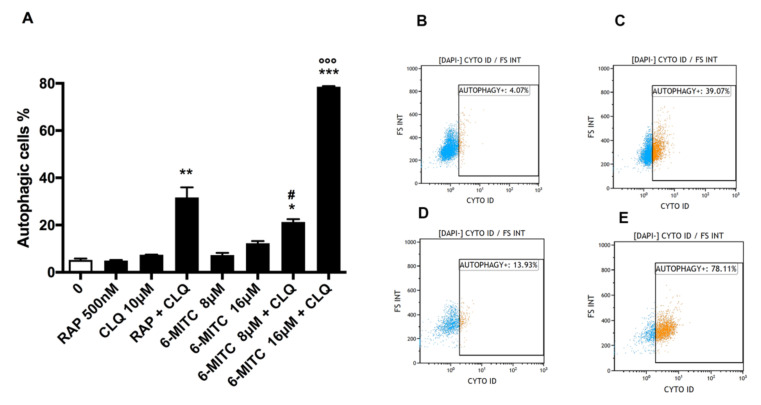
Percentage of autophagic cells in HL-60 cells after 24 h treatment with 6-MITC and/or positive control RAP and/or lysosomal inhibitor CLQ at the indicated concentrations compared to the concurrent negative control cultures (0 μM) (**A**). Each bar represents the mean ± SEM of three independent experiments followed by the Dunnet post-test. * *p* < 0.05 vs. control ** *p* < 0.01 vs. control; *** *p* < 0.001 vs. control; # *p* < 0.05 vs. 6-MITC 8 μM; °°° *p* < 0.001 vs. 6-MITC 16 μM. Representative FCM plots of 6-MITC 0 μM (**B**), RAP + CLQ (**C**), 6-MITC 16 μM (**D**) and 6-MITC 16 μM + CLQ (**E**).

**Figure 3 biomolecules-12-01485-f003:**
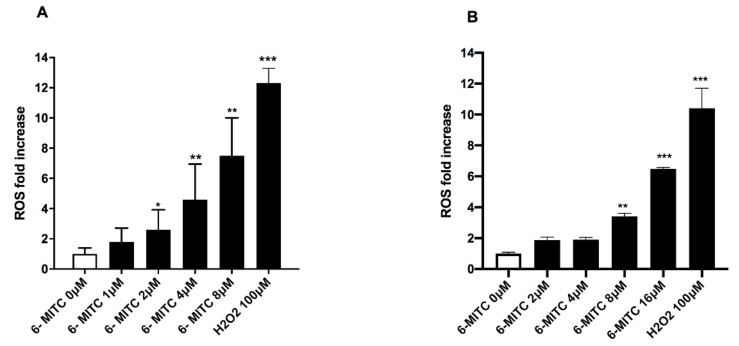
ROS fold increase on Jurkat (**A**) and HL-60 (**B**) cells after 24 h treatment with 6-MITC at the indicated concentrations compared to the concurrent negative control cultures (0 μM). H_2_O_2_ was used as a positive control. Each bar represents the mean ± SEM of three independent experiments followed by the Dunnet post-test. * *p* < 0.05 vs. control; ** *p* < 0.01 vs. control; *** *p* < 0.001 vs. control.

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
