# Peer review of "6-(Methylsulfonyl) Hexyl Isothiocyanate: A Chemopreventive Agent Inducing Autophagy in Leukemia Cell Lines"

_biomolecules, 2022, doi:10.3390/biom12101485_

Round 1

Reviewer 1 Report

Dear authors,

In this study, the authors attempted to evaluate whether 6-MITC induced Jurkat and HL-60 cells autophagy via inducing oxidative stress in the cells. In my opinion, this is an interest study for illustrating the biological function of 6-MITC. I have several suggestion and question about the experimental design and writing to improve this study:

1. Did the authors try another autophagy inhibitors during the study? In both figure 1 & 2, we can clearly observed that chloroquine potentiate the autophagy cell population in Jurkat and HL-60 cell lines while co-incubating with rapamycin or 6-MITC. This result means that chloroquine is not a proper autophagy inhibitor for this study. I strongly suggest the authors test combination of 6-MITC with other autophagy inhibitors such as 3-methyladenine or bafilomycin A1 to ensure the authors' claim.

2. Was autophagy one of the major contributor of the cytotoxicity of 6-MITC? The overall effect of autophagy can be various, including cell death, cytostatic effect, transformation, senescence, and etc, and which can be major or minor. I suggest the authors evaluating the cytotoxicity of 6-MITC against Jurkat and HL-60 combining with autophagy inhibitors.

3. The authors need to test the dynamic of autophagy cell population in combination of 6-MITC and ROS inhibitors such as N-acetylcysteine, ascorbic acid, or alpha-tocopherol.  The reason is the same with the above suggestion.

4. From the introduction of this study, we can know that 6-MITC is an antioxidant (Line 81-82). However, in the Figure 3, the authors demonstrated that 6-MITC induced oxidative stress in the tumor cells. How did this contradicted result happen? The authors need to discuss this.

5. For the introduction, I suggest the authors introducing the markers about the autophagy and describing more about the known biological activity or tested models of 6-MITC. The current introduction looks like a fundamental introduction of autophagy.

6. Similar issue can be observed in the discussion. The authors need to discuss more about the biological details about 6-MITC, such as the known action mechanisms and biological functions. The current discussion discussed autophagy too much.

7. A minor issue: on Line 136, the chloroquine is negative control instead of positive control.

Author Response

We would like to thank the Reviewer for their comments and suggestions, which have been very helpful in improving the quality of the manuscript.

All Reviewer’s comments and suggestions have been addressed and the changes made are listed point by point and highlighted in red in the manuscript for quicker viewing.

1. Did the authors try another autophagy inhibitors during the study? In both figure 1 & 2, we can clearly observe that chloroquine potentiates the autophagy cell population in Jurkat and HL-60 cell lines while co-incubating with rapamycin or 6-MITC. This result means that chloroquine is not a proper autophagy inhibitor for this study. I strongly suggest the authors test combination of 6-MITC with other autophagy inhibitors such as 3-methyladenine or bafilomycin A1 to ensure the authors' claim.

We thank the reviewer for the observation, which allowed us to clarify this concept. Chloroquine (CLQ) was used as an inhibitor of the autophagic vesicles degradation, since it acts as lysosomal inhibitor by raising the lysosomal pH and ultimately inhibits the fusion between autophagosomes and lysosomes, thus preventing the maturation of autophagosomes into autolysosomes; as a result, CLQ blocks a late step of autophagy leading to an accumulation of autophagic vesicles in the cell and therefore to an augmented signal. We chose to use it in our experiments to enhance the autophagy signals, otherwise low due to the fast clearance of autophagic vesicles, as reported in “Guidelines for the use and interpretation of assays for monitoring autophagy (4th edition)” doi: 10.1080/15548627.2020.1797280.

We clarify this aspect in Materials and Methods, Results and Discussion sections.

2. Was autophagy one of the major contributors of the cytotoxicity of 6-MITC? The overall effect of autophagy can be various, including cell death, cytostatic effect, transformation, senescence, and etc, and which can be major or minor. I suggest the authors evaluating the cytotoxicity of 6-MITC against Jurkat and HL-60 combining with autophagy inhibitors.

The question is certainly rightful and of interest but the quantification of the autophagy contribution to cytotoxicity is not an easy task. Therefore, as suggested by the reviewer, we evaluated the cytotoxicity of 6-MITC against Jurkat and HL-60 combining it with the autophagy inhibitor CLQ and evaluating cytotoxicity in terms of viable, necrotic and apoptotic cells by the double staining 7AAD/Annexin-V-PE. However, the results obtained did not permit to evidence any significant change in the cytotoxic profile of 6-MITC as reported in the Results and Discussion section.

3. The authors need to test the dynamic of autophagy cell population in combination of 6-MITC and ROS inhibitors such as N-acetylcysteine, ascorbic acid, or alpha-tocopherol. The reason is the same with the above suggestion.

We thank the reviewer for the rightful observation. As suggested, in order to test the dynamic of the autophagic process in the cell population we co-treated with a combination of 6-MITC and the antioxidant N-acetylcysteine (NAC) HL60 and Jurkat cells for 24h. The results showed a decrease in the CYTO-ID mean green fluorescence of the cell population co-treated with 6-MITC and NAC compared to that of the cell population treated only with 6-MITC (figure 1S supplementary materials). This outcome allows to hypothesize that oxidative stress could be involved in the modulation of autophagy induced by 6-MITC; however, it represents an extremely preliminary experiment which will require more in-depth analyses to be confirmed. The possible link between ROS and autophagy induction by 6-MITC has been dealt with in Materials and Methods, Results and Discussion sections.

4. From the introduction of this study, we can know that 6-MITC is an antioxidant (Line 81-82). However, in the Figure 3, the authors demonstrated that 6-MITC induced oxidative stress in the tumor cells. How did this contradicted result happen? The authors need to discuss this. 

We dealt with this aspect in the Discussion section.

5. For the introduction, I suggest the authors introducing the markers about the autophagy and describing more about the known biological activity or tested models of 6-MITC. The current introduction looks like a fundamental introduction of autophagy.

We improved the Introduction section as suggested.

6. Similar issue can be observed in the discussion. The authors need to discuss more about the biological details about 6-MITC, such as the known action mechanisms and biological functions. The current discussion discussed autophagy too much.

We improved the Discussion section as suggested.

7. A minor issue: on Line 136, the chloroquine is negative control instead of positive control.

We thank the reviewer for the rightful observation. We reported CLQ as a positive control as described in the employed autophagy kit manual due to the fact that CLQ is used to increase autophagic cells signals. Understanding, though, that calling CLQ a positive control could be misleading, we better explained CLQ activity and its use in Materials and Methods, Results and the Discussion sections.

Reviewer 2 Report

The authors have focused on autophagy and ROS to analyze the mechanism of the antitumor effects of 6-MITC, which exhibits cytotoxic, growth inhibitory, and differentiation-inducing effects on leukemia cells. However, it has already been reported that 6-MITC induces autophagy in human leukemia cells K562 in a concentration-dependent manner, as indicated by detection of autophagosomes by electron microscopy and conversion to LCII by western blotting (#41 or Biomoplecules 2019, 9, 774). Therefore, the authors' claim that 6-MITC induces autophagy in human leukemia Jarkat and HL-60 cells is not novel. Moreover, there may be methodological problems with the quantitative nature of the autophagy detection method. (Figure 1B-E, The window setting for detecting autophagy cells appears to be arbitrary. In particular, the window in panel B is clearly different from the other three panels (C, D, and E).)

If there is a causal relationship between increased ROS production and autophagy induction, it should be shown experimentally. Also, if ROS is triggering autophagy, a hypothesis regarding the mechanism by which 6-MITC, with its known antioxidant effect, causes increased ROS production should be presented.

In a previous study (#30 or Oncotarget 2017, 8,111697), the authors reported that 6-MITC induces CASPASE-8-mediated apoptosis in Jarkat and HL-60 cells, but in this paper the authors did not In this paper, the authors need to clarify the relationship between apoptosis and autophagy in terms of cell death induction.

Author Response

We would like to thank the Reviewer for their comments and suggestions, which have been very helpful in improving the quality of the manuscript.

All Reviewer’s comments and suggestions have been addressed and the changes made are listed point by point and highlighted in red in the manuscript for quicker viewing.

1. The authors have focused on autophagy and ROS to analyze the mechanism of the antitumor effects of 6-MITC, which exhibits cytotoxic, growth inhibitory, and differentiation-inducing effects on leukemia cells. However, it has already been reported that 6-MITC induces autophagy in human leukemia cells K562 in a concentration-dependent manner, as indicated by detection of autophagosomes by electron microscopy and conversion to LCII by western blotting (#41 or Biomolecules 2019, 9, 774). Therefore, the authors' claim that 6-MITC induces autophagy in human leukemia Jarkat and HL-60 cells is not novel. Moreover, there may be methodological problems with the quantitative nature of the autophagy detection method. (Figure 1B-E, The window setting for detecting autophagy cells appears to be arbitrary. In particular, the window in panel B is clearly different from the other three panels (C, D, and E).)

We thank the reviewer for the suggestions and observations. The above-mentioned article, it was already cited in the Discussion section, but now we included it also in the Introduction section and highlighted this aspect. As concerns methodological problems with the quantitative nature of the autophagy detection method, we thank the reviewer for noticing the mistake in figure 1. We wrongly included an incorrect version of panel B. Now we have replaced it with its correct version.

2. If there is a causal relationship between increased ROS production and autophagy induction, it should be shown experimentally. Also, if ROS is triggering autophagy, a hypothesis regarding the mechanism by which 6-MITC, with its known antioxidant effect, causes increased ROS production should be presented.

We thank the reviewer for the rightful suggestion. As suggested also by reviewer 1, in order to test the dynamic of the autophagic process in the cell population we cotreated HL60 and Jurkat cells with a combination of 6-MITC and the antioxidant N-acetylcysteine (NAC) for 24h. The results showed a decrease in the CYTO-ID mean green fluorescence of the cell population cotreated with 6-MITC and NAC compared to that of the cell population treated only with 6-MITC (figure 1S supplementary materials). This outcome allows to hypothesize that oxidative stress could be involved in the modulation of autophagy induced by 6-MITC; however, it represents an extremely preliminary experiment which will require more in-depth analyses to be confirmed. The possible link between ROS and autophagy induction by 6-MITC has been dealt with in Materials and Methods, Results and Discussion sections.

3. In a previous study (#30 or Oncotarget 2017, 8,111697), the authors reported that 6-MITC induces CASPASE-8-mediated apoptosis in Jurkat and HL-60 cells, but in this paper the authors did not In this paper, the authors need to clarify the relationship between apoptosis and autophagy in terms of cell death induction. 

As suggested, we improved the Discussion section also regarding this aspect. The quantification of the autophagy contribution to cytotoxicity is not an easy task. So, following the suggestion of reviewer 1, we evaluated the cytotoxicity of 6-MITC against Jurkat and HL-60 combining it with the autophagy inhibitor CLQ and evaluating cytotoxicity in terms of viable, necrotic and apoptotic cells by the double staining 7AAD/Annexin-V-PE. However, the results obtained did not permit to evidence any significant change in the cytotoxic profile of 6-MITC, as reported in Results and Discussion section.

Round 2

Reviewer 1 Report

Dear authors,

I thank you for your detailed explanation for your study. I have no further concern to your study.

Reviewer 2 Report

Very well revised. I have no more comment.